# Cartilage Formation In Vivo Using High Concentration Collagen-Based Bioink with MSC and Decellularized ECM Granules

**DOI:** 10.3390/ijms23052703

**Published:** 2022-02-28

**Authors:** Elena V. Isaeva, Evgeny E. Beketov, Grigory A. Demyashkin, Nina D. Yakovleva, Nadezhda V. Arguchinskaya, Anastas A. Kisel, Tatiana S. Lagoda, Egor P. Malakhov, Anna N. Smirnova, Vasiliy M. Petriev, Petr S. Eremin, Egor O. Osidak, Sergey P. Domogatsky, Sergey A. Ivanov, Petr V. Shegay, Andrey D. Kaprin

**Affiliations:** 1A Tsyb Medical Radiological Research Centre, Branch of the National Medical Research Radiological Centre of the Ministry of Health of the Russian Federation, 249036 Obninsk, Russia; kusimona@yandex.ru (E.V.I.); dr.dga@mail.ru (G.A.D.); yakovleva.40@mail.ru (N.D.Y.); nnv1994@mail.ru (N.V.A.); ki7el@mail.ru (A.A.K.); la-goda@yandex.ru (T.S.L.); malaxowegor@yandex.ru (E.P.M.); filimonowa.af@gmail.com (A.N.S.); petriev@mrrc.obninsk.ru (V.M.P.); oncourolog@gmail.com (S.A.I.); 2Institute for Nuclear Power Engineering, National Research Nuclear University Mephi Obninsk, 249039 Obninsk, Russia; 3Federal State Autonomous Educational Institution of Higher Education I.M. Sechenov, First Moscow State Medical University of the Ministry of Health of the Russian Federation, 119991 Moscow, Russia; 4National Medical Research Center of Rehabilitation and Balneology of Ministry of Healthcare of the Russian Federation, Federal State Budgetary Institution, 121099 Moscow, Russia; ereminps@gmail.com; 5Imtek Ltd., 121552 Moscow, Russia; eosidak@gmail.com (E.O.O.); spdomo@gmail.com (S.P.D.); 6Federal State Budgetary Institution, National Medical Research Centre of Cardiology, Ministry of Health of the Russian Federation, 121552 Moscow, Russia; 7National Medical Research Radiological Centre of the Ministry of Health of the Russian Federation, 249036 Obninsk, Russia; dr.shegai@mail.ru (P.V.S.); kaprin@mail.ru (A.D.K.)

**Keywords:** bioprinting, collagen, bioink, MSC, ECM, cartilage

## Abstract

The aim of this study was to verify the applicability of high-concentration collagen-based bioink with MSC (ADSC) and decellularized ECM granules for the formation of cartilage tissue de novo after subcutaneous implantation of the scaffolds in rats. The printability of the bioink (4% collagen, 2.5% decellularized ECM granules, derived via 280 μm sieve) was shown. Three collagen-based compositions were studied: (1) with ECM; (2) with MSC; (3) with ECM and MSC. It has been established that decellularized ECM granules are able to stimulate chondrogenesis both in cell-free and MSC-laden scaffolds. Undesirable effects have been identified: bone formation as well as cartilage formation outside of the scaffold area. The key perspectives and limitations of ECM granules (powder) application have been discussed.

## 1. Introduction

Extrusion-based bioprinting is a key technique of tissue engineering and biofabrication. It can be used to form scaffolds of a certain external shape and complex internal structure [1,2]. Bioprinting is fully dependent on bioinks, which include biomaterials in a form of hydrogels, growth factors (or other signaling molecules) and cells. The development of new biomaterials, in general, and bioinks in particular, is a long-term process. The materials (as well as prepared compositions) must ensure certain scaffold mechanical properties and conditions for cell survival, proliferation, a strictly defined effect on cell phenotype, adequate and predictable biodegradation rates, etc. [3,4]. The bioink development process involves these properties testing both in vitro and in vivo. Cartilage could be considered as a relatively simple tissue that has no blood and lymphatic vessels, nor nerves. The tissue has low cell density (1–5% of the total cartilage volume). All this makes cartilage an appropriate model for these purposes [5,6].

There are four main strategies related to cells in a bioink. The first one refers to the use of cell-free scaffolds; the other three imply usage of cells: differentiated cells of a certain type of tissue (in the case of cartilage, chondroblasts, and chondrocytes), a mixture of differentiated cells and MSC (differentiated cells serve as a source of growth factors), or only MSC (with the mandatory use of growth factors) [7,8]. The disadvantage of differentiated cells (of almost any tissue, including cartilage) is related to the rapid loss of a cell’s specific phenotype in the course of its cultivation, aimed at obtaining the required amount [9,10]. An alternative source of chondrogenic progenitor cells can be adult stem cells obtained from various tissues (adipose, bone marrow, dental pulp, etc.) [11,12,13]. Adipose tissue is a well-known depot of multipotent adult stem cells that are readily available and that can be obtained in large quantities using relatively uncomplicated invasive procedures, such as lipoaspiration or liposuction [14,15,16].

During cultivation, adipose-derived stem cells (ADSC) are able to maintain a stable undifferentiated phenotype without changing telomerase activity for nine passages, thus, acting as an abundant source of cells without losing multipotency [14]. They also have the ability to differentiate into chondrocytes in response to specific environmental signals [17,18,19]. However, the use of exogenous growth factors in clinical practice is not always feasible due to its cost or regulatory requirements. Thus, bioactive scaffold that provides appropriate environmental signals represents an alternative approach for inducing ADSC chondrogenesis. This function could hardly be performed by a main bioink biomaterial, since most of the materials, such as collagen, gelatin, agarose, silk fibroin, sodium alginate, etc., are not able to recreate the microenvironment that is typical for cells in a tissue, nor for the internal morphology and function of cartilage [3,20]. The extracellular matrix (ECM) itself can be an ideal example of such a material. The composition of ECM of each tissue is unique. It is a complex network of components, the majority of which are fiber-forming proteins such as collagens, elastin, fibronectin, laminins, glycoproteins, proteoglycans, and glycosaminoglycans (GAG) [21]. ECM regulates many cellular processes, including growth, migration, cell differentiation, homeostasis, and morphogenesis [21,22]. It is the main source and conductor of biochemical and biomechanical signals, ensuring organization and functioning of the tissue proper [23], providing an appropriate microenvironment for the cells [24]. The importance of using biomaterials, including dECMs, in regenerative medicine can hardly be overestimated, since they allow you to restore the function of the organ, and not just its volume. Thus, [25] developed a decellularized dental pulp (DDP) matrix loaded by human dental pulp stem cells (hDPSCs) in co-treatment with extracellular vesicles (EVs), which might enhance the dentinogenic differentiation with a high potentiality for endodontic regeneration.

The ability of porous, cartilage ECM-based scaffolds to induce ADSC chondrogenic differentiation without exogenous growth factors has already been described [13,17]. There are data on the use of hydrogels based on decellularized ECM (dECM) via bioprinting. Bioinks based on dECM initially possess tissue-specific properties. The data are available for various types of tissues: liver, heart, adipose and cartilaginous tissues are currently being investigated [26,27,28,29,30,31]. However, the use of dECM-based hydrogel is limited by weak mechanical properties and low printing accuracy, as well as by lack of customizable physical properties. For these reasons, the hydrogel is often used in combination with other materials, such as gelatin methacryloyl [32], fibrin [33], nanocellulose [34], silk fibroin [35], or cross-linking agents, genipin [36], *N*-(3-Dimethylaminopropyl)-*N*'-ethylcarbodiimide hydrochloride [36], and riboflavin [37]. At the same time, an attempt to improve both mechanical properties of a scaffold and printability of the bioink also affects scaffold elasticity. 

The granules of dECM (powder) slightly affect (improve) rheological properties and eliminate the use of any toxic photo- or chemical cross-linking agents. Previously, dECM was used in the form of microparticles in a hydrogel based on gelatin and fibrinogen and silk fibroin, respectively [38,39]. Promising results from cytocompatibility studies have been obtained.

The aim of this study was to verify the applicability of high-concentration collagen-based bioink with MSC (ADSC) and decellularized ECM granules for the formation of cartilage tissue de novo after subcutaneous implantation of the scaffolds in rats. The study design is shown in Figure 1.

## 2. Results

### 2.1. dECM Tissue

Costal cartilage before (Figure 2a) and after (Figure 2b,c) decellularization. Certain changes in the tissue are evident. Empty gaps, in which isogenic groups of cells were located, are clearly visible under light microscopy. The cellular and fibrous layers of the perichondrium are preserved. An intensive staining with alcian blue of the extracellular matrix of decellularized cartilage showed the preservation of GAG molecules (Figure 2d). Thus, the decellularization method used in the study allows for eliminating the cells while preserving the perichondrium. Decellularized samples of cartilage tissue were used to obtain dECM granules.

### 2.2. dECM Granules

After, the lyophilization and homogenization dECM granules diameter was measured. Its probability distribution is shown in Figure 3. The diameter median and interquartile range (ICR) was 22.7 ÷ 155.5 µm with a median of 42.8 µm. The results included the effect of conglomerate forming that occurred both during its storage and in the course of sifting. The effect explains a significant number of granules whose diameter exceeded the sieve size (280 µm). Its quota was 10%. The analyzed dECM granules were used for the study.

### 2.3. Bioink Printability

Since dECM granules (2.5%) were added to the bioink (based on 4% collagen), the hydrogel suitability for extrusion-based printing had to be verified. For this purpose, test printing was conducted. The object is shown in Figure 4. It helped to check both filament extrusion stability and printing accuracy. The test printing allowed for estimating the effect of material fluidity, swelling and polymerization. Two parameters were taken into account: (1) line thickness (3 layers in height) in 12 places and the area of the formed niches in eight places. Both parameters were evaluated in dynamics: immediately after printing and in 24 h (after incubation at +37 °C). The results of such a testing are shown in Table 1.

Extrusion of the material through a 21G (~514 μm) needle was stable during the testing. Thus, 4% collagen hydrogel containing 2.5% dECM with a particle size of up to 280 μm (IQR = 22.7 ÷ 155.5 µm) was found to be suitable for extrusion-based bioprinting with the used printing parameters. Analysis of Table 1 data indicates that filament tended to swell during the incubation. It effected not only filament thickness, but also, as expected, it decreased the volume of the formed niches. This effect is characteristic of collagen, as well as of any hydrogel [40]. According to the values of the confidence intervals (both in the case of the filament thickness and the niche area), dECM granules increased the swelling. However, the effect was statistically significant only in the case of niches after incubation (the minimum difference was 0.029 mm^2^, which corresponded to 2% of the niche area for collagen after incubation).

In general, the swelling led to a thickening of the line (filament in 3 layers) after the incubation in 1.75 times in the case of collagen and in 1.86 times in the case of the collagen with dECM. This was taken into account in the course of subsequent printing sessions.

### 2.4. ADSC Phenotype

The cells had a regular fusiform shape with 2–4 long processes; the cytoplasm was homogeneous and transparent, without any inclusions (Figure 5). The nuclei were located closer to the periphery (eccentrically) with a uniform distribution of chromatin. The cell culture had a high proliferative potential. The cell population doubling time was 24.31 ± 2.19 h.

The results (Table 2) of the immunophenotypic study indicate that the cells used in the study strongly belonged to the MSC group.

### 2.5. Scaffold Implantation

The results obtained indicate the dynamics in cell’s chondrogenic differentiation. The most pronounced effect was observed in the 3rd group, where a full component bioink was used; the least evident result was in the 2nd group. The main adverse effect—ossification—was revealed in the case of all three types of the scaffold. Along with cartilage and bone, brown adipose tissue (BAT) formation was detected. Inflammatory reactions were weak. There is a description related to each experimental group.

#### 2.5.1. 1st Group (Collagen + dECM)

In one week after the implantation, a connective tissue capsule with a moderate content of cellular and fibrous structures was observed around the scaffolds. Cavities filled with fibrillar material were detected in the implant (Figure 6). The presence of an inflammatory reaction with monocytes, lymphocytes and a small number of multinucleated resorption cells was noted. In the subcutaneous adipose tissue, groups of multilocular lipocytes (BAT cells) were found, characterized by an intense positive reaction to PCNA (Figure 7) and PCNA-positive multinucleated cells of foreign body resorption in the connective tissue capsule around the scaffold (Figure 6). In both animals of this group, euthanized in one week after implantation, among the striated muscle tissue near the scaffold, the formation of islets of cartilaginous tissue bound by the perichondrium was noted. Cartilage cells (apparently chondroblasts) were in capsules, characterized by a large round nucleus, a small basophilic cytoplasm, the absence of a territorial matrix and isogenic cell groups (Figure 8(A1)). The nuclei of these cells were intensely stained for PCNA (Figure 8(B1)); the cytoplasm gave an intense reaction for type II collagen (Figure 8(C1)) The intercellular substance was stained with alcian blue (Figure 8(D1)). Thus, GAG (the most specific product of cartilage) were detected. In one animal, along with cartilaginous tissue (Figure 9a), formation of bone tissue was also found (Figure 9b).

In two weeks, a well-vascularized connective tissue capsule was formed around the implant (Figure 10a,b). At this time-point, an intensive resorption of the scaffold material was observed corresponding to multinucleated resorption cells (Figure 11a,b). Type II collagen was detected in the cytoplasm of these cells, as well as macrophages present in the connective tissue capsule (Figure 12a,b). In the subcutaneous connective tissue around the striated muscle fibers (as well as in the 1-week period), in addition to the expected white adipose tissue, BAT was also observed. It came from multilocular lipocytes that gave a distinct reaction to PCNA (Figure 13). One animal (from two) did not have any evidence of cartilaginous tissue formation.

#### 2.5.2. 2nd Group (Collagen + MSC)

On the 7th day after the implantation, a connective tissue capsule was formed around the scaffolds. In one of two animals, near the scaffold and muscle fibers, the formation of unmineralized bone plates (osteoids) with a large number of osteoblasts (randomly located along the periphery of the osteoids) were observed (Figure 14a,b).

In two weeks, the formation of a powerful connective tissue capsule with numerous blood vessels was observed. It grew into the scaffold, dividing it into separate fragments (Figure 15a,b). In the connective tissue capsule along the periphery of the implant, there were multinucleated resorption cells, the presence of an inflammatory infiltrate containing a large number of eosinophilic leukocytes (Figure 16). The cells with cytoplasm containing type II collagen was detected (Figure 17a,b). It is highly probable that these cells were macrophages that captured the scaffold material. A significant number of mast cells were detected in the connective tissue capsule, the cytoplasm of which was selectively stained with alcian blue (since the reaction was performed in an acidic environment). Deep in the connective tissue capsule, fragments of the scaffold and necrotic tissues surrounded by multi-layered epithelium were found. The growth of BAT was noted. In the same animal, a small island of the irregular shape of cartilaginous tissue was found near the scaffold (Figure 8(A2–D2)). That was the only evidence of cartilage formed in the group.

#### 2.5.3. 3rd Group (Collagen + dECM + MSC)

In animals of this group, in one week after implantation, the formation of a powerful connective tissue capsule was also noted. It (similar to the 2nd group) invaded into the implant and divided it into large fragments (Figure 18a,b). In the connective tissue capsule (on the border with the implant), giant resorption cells were detected. In one animal, a cord of cartilage cells surrounded by a perichondrium was found. At the same time, there were morphological signs of indirect osteogenesis as well (Figure 19). In another animal near the implant, among the group of striated muscle fibers, a cartilage tissue was revealed. It contained chondroblasts located in capsules (Figure 8(A3–D3)).

On the 14th day after the implantation cartilage tissue were well observed (Figure 20). Chondroblasts formed small isogenic groups and an extracellular matrix with a pronounced perichondrium (Figure 8(A3), Figure 20). The tissue gave an intense reaction to collagen type II (Figure 8(B4)). It showed signs of a decrease in proliferative activity (Figure 8(C4)). Its extracellular matrix had accumulated a large amount of GAG since it was intensely stained with alcian blue (Figure 8(D4)). According to immunohistochemical staining, the proliferative activity of connective tissue cells and multinucleated cells of foreign body resorption in the scaffold remained high (Figure 21).

## 3. Discussion

ECM of native cartilage is a dense connective tissue that includes a highly organized network of collagen (mainly type II) and large aggregating proteoglycans (e.g., aggrecans) [13,41]. Achievement of full decellularization of native cartilage is an urgent problem because the density of ECM makes a complete process difficult due to limited diffusion of reagents [42]. The decellularization protocol used in the study allowed us to remove almost all cellular elements from the costal cartilage tissue, with the exception of the perichondrium. In order to increase the efficiency of chemical decellularization, tissue is often mechanically destroyed, but this changes mechanical properties of the matrix [43]. In addition, we were interested in the maximum retention of signaling molecules responsible for cell differentiation, since the objectives of the study did not imply the use of growth factors or other stimulants of chondrogenesis. There are other studies in which, instead of dECM, a powder of native lyophilized cartilage was used to compose a bioink [33]. Decellularization reduces the immunogenicity of the matrix by removing cellular antigens, which, theoretically, tend to induce an immune response [44]. This is true primarily for xenogenic matrix sources. On the other hand, the molecules that make up ECM are highly conservative in all mammalian species; xenogenic effect (for cell-free ECM) is not so obvious [45]. In our research, scaffolds had mixed nature: collagen (pig—xenogenic), dECM (rat—allogenic), MSC (human—xenogenic). Histological examination revealed a small inflammatory reaction in animals in the experiment, including the group where the scaffold did not contain dECM. Moreover, the reaction was more pronounced when MSC was used. In our opinion, it can be both a consequence of surgical intervention and the use of human MSC (xenogeneic source) rather than dECM and even less by type I collagen.

Decellularized ECM is used for tissue engineering, mostly in the form of a gel, which is obtained after enzymatic dissolution of the powder under acidic conditions. There are few works that describe the use of the powder directly as functional addition. In two of them [46,47], fresh cartilage particles or dECM obtained from articular cartilage were used as microcarriers for the cultivation of bone marrow stromal cells (BMSC) and chondrocytes. Moreover, if, in the work of J. Barthold et al. [46], dECM powder was added to hyaluronic acid-based hydrogel and cell migration to the granules was observed, then, in the study conducted by H. Yin et al. [47], BMSC differentiated into mature chondrocytes was achieved in 21 days without the use of exogenous growth factors. In another study [48], human periosteal cell and demineralized bone granules (250–500 μm) were successfully used to induce the cells’ osteoinductive potential without any growth factors in the collagen-based scaffold. The effect of ECM powder (non-decellularized) was shown for ADSM-laden fibrin-based scaffold [33]. It was also revealed that the ECM powder from the synovial membrane has the potential to stimulate the expression of the type II collagen gene [49]. A few works report the use of non-decellularized ECM powder as a part of a bioink, e.g., for articular cartilage [39] or liver tissue [38].

The amount (concentration) of ECM, as well as its granules size and morphology, must be taken into account among a number of other factors [50]. In our research, dECM concentration was 2.5% (*w*/*v*). Usually, the amount of ECM varies from 1 to 20%. The rule “the higher the better” does not always work, since the supplement strongly affects hydrogel behavior. It was shown for injectable fibrin-based hydrogel [33], where two ECM granules (97 ± 26 μm) concentrations were used: 2 and 10%. Unexpectedly, the higher ECM amount impeded gel formation.

In addition to the concentration of the powder, it is necessary to take into account the viscosity of base hydrogel, since the uniformity of the distribution of granules within the volume of bioink (and, thus, the scaffold) and their possible adhesion with the formation of conglomerates (that could clog the needle) depend on it [38]. In our research, we did not face any problem related to printability of the bioink. The effect of uneven distribution of the granules within the scaffold volume was not studied. The uneven distribution of the granules can lead to an inhomogeneous formation of tissue. Thus, the issue should be considered as important.

The granules (powder as well) retain the micro- and ultrastructure of the original ECM but have a significantly increased surface area available for interaction with cells [45]. Moreover, the geometric shape of ECM particles is a decisive factor in determining its suitability as a substrate for adhesion of attachment-dependent cells, their migration, proliferation, and differentiation in vivo [51]. However, the data on the proper size of the particles are contradictory. Large granules (from 250 µm) can function as microcarriers [46,47]. From the point of view of bioink homogeneity and better printability, the particles should be smaller: no more than 50 µm [38]. It was shown that dECM particles with a size of about 50 µm can enhance the differentiation of cells in the chondrogenic direction [52]; chondroinductive ability was revealed in the case of 97 µm [33]. In turn, the particles with a diameter of 52 µm or less undergo phagocytosis in the body and disrupt the function of the host's phagocytes [53]. Based on this, we used granules with a diameter of a broad range: from a powder state to 280 μm. The upper limit was determined by the inner diameter of the needle (514 µm) since the granules should not cause clogging in case of swelling during the printing. In accordance with our results, granules with a diameter up to 280 μm (0.9 quantile) did not serve as microcarriers, but its amount (2.5%) was enough to stimulate chondrogenesis.

There are several assumptions of the mechanism by which cartilage ECM induce chondrogenesis in vitro. ECM particles themselves are potentially chondroinductive due to the content of GAG, such as chondroitin sulfate and aggrecan [44,54], collagen and growth factors, especially from the TGF-β superfamily [13,33,44]. It is known that the presence of type II collagen in ECM is beneficial for chondrogenesis [55,56]. The specific size of the ECM particles is also a factor that may enhance chondrogenesis in vitro and in vivo [33]. The mechanisms by which ECM-containing scaffolds promote tissue remodeling in the body include mechanical support, degradation and release of bioactive molecules, recruitment and differentiation of endogenous stem cells (progenitor cells), and modulation of the immune response to an anti-inflammatory phenotype [45].

The results of our experiments showed that the dECM granules’ presence in the scaffold provide it with the necessary signaling molecules. We observed chondrogenesis both in the 1st group of animals, where we use scaffolds with only dECM, and in the 3rd group, where tissue-engineered constructions contained both dECM and stem cells. Only one of the animals of the 2nd group, whose scaffolds contained MSC without ECM, formed a small island of cartilaginous tissue in two weeks after the implantation. It should be noted that, in almost all cases, islets of cartilaginous tissue were formed not in the scaffold itself, but nearby, among the muscle fibers. It is known that muscles are well supplied with blood. Thus, our study again raises the problem of vascularization of tissue-engineered structures [57,58]. Even in the case of cartilage (generally non-vascularized tissue), the issue is urgent.

Another open question, which was discussed earlier [59], corresponds to the necessity of the MSC presence in the scaffolds. From the point of view of tissue engineering, the production of cell-free scaffolds is preferable, since bioprinting with cells imposes a number of restrictions on the process. The immunogenicity of cell-free constructs will also be lower, unless the patient's own cells are used. In all groups, in one of the two animals euthanized after one week, we observed osteogenesis with the formation of coarse fibrous bone tissue, which occurs mainly in the embryos. We also observed bone formation, in addition to cartilaginous tissue, in our previous study, after subcutaneous implantation of scaffolds containing 4% type I collagen and rat chondrocytes in rats [60]. This is probably due to the peculiarities of the molecular mechanisms of regulation of osteo- and chondrogenesis in specific animals, since MSC derived from adipose tissue are equally capable of differentiating into cells such as osteoblasts, chondroblasts, and adipocytes [61]. Perhaps this also explains the appearance of BAT under the skin, which is characteristic mainly of newborns. Since all these phenomena were observed to one degree or another in all groups, another question that arises is whether we can somehow regulate the processes of stem cell differentiation in the body to prevent such undesirable reactions. For a final clarification of all these questions, the study must be continued.

## 4. Materials and Methods

### 4.1. MSC Culture

The study was performed using MSC isolated from human adipose tissue. Women (*n* = 10) aged 25 to 40 were selected as donors. The donation was carried out during elective liposuction operations after patients signed informed consent. Isolation of cells was conducted by enzymatic treatment of adipose tissue with 0.015% type II collagenase solution (Sigma-Aldrich, St. Louis, MO, USA) according to the previously described method [62]. Briefly, the lipoaspirate was washed 3 times from the tumescent fluid using Hartman’s solution (Biochemist, Moscow, Russia), after which it was treated with collagenase for 30 min at +37 °C. At the end of the treatment, Hartman’s solution was added (1:1) to reduce the enzyme activity. The resulting cell suspension was filtered (100 μm, SPL, Pocheon, South Korea) and washed 3 times with PBS (hereinafter, unless otherwise stated—PanEco, Moscow, Russia).

The primary cell culture was transferred into a culture flask (75 cm^2^, Corning, New York, NY, USA) at a rate of 5 × 10^5^ cells per cm^2^. The cells were cultured according to the standard procedure [63] up to the 5th passage at +37 °C in 5% CO_2_ atmosphere. Medium composition: 1 g/L glucose content DMEM medium, 10% fetal bovine serum (Biosera, Nuaille, France), penicillin-streptomycin (100 U/mL and μg/mL, respectively), glutamine (150 μg/mL). Cells were examined and photographed using Biomed 3 microscope (Biomed, Moscow, Russia) with ToupCam 3.1 camera (ToupTek, Hangzhou, China). In order to confirm that the isolated cells belonged to ADSC, the phenotypic profile was studied using a BD FACS Canto II flow cytometer (Becton Dickinson, Franklin Lakes, NJ, USA) for the main surface markers: CD13(+), CD34(−), CD44(+), CD45(−), CD73(+), CD90(+), CD105(+), CD146(−) (Becton Dickinson, Franklin Lakes, NJ, USA).

### 4.2. dECM Granules

ECM was obtained from costal cartilage of 12 female Wistar rats aged from 1 to 1.5 months. The procedures described below were conducted under the permission of the Bioethical Commission on Keeping and Using Laboratory Animals of A. Tsyb MRRC No. 1-N-00007 dated 7 May 2021. Decellularization was performed according to a modified protocol of the detergent-enzymatic method [64]. After euthanasia, ribs were removed from animals under sterile conditions, soft tissues were mechanically removed, and cartilage was excised. The costal cartilage was stirred in a mixture (1:1) of 0.25% trypsin and 0.2% type I collagenase solution (Gibco, Waltham, MA, USA) for 40 min at +37 °C in order to remove the remnants of soft tissues. The procedure was repeated twice, then the perfusion was performed sequentially: (1) PBS, 30 min; (2) 0.05% EDTA (Chemmed, Laverna, Russia), 24 h; (3) deionized water, 30 min; (4) 3% SDS (Panreac, Castellar del Vallès Barcelona), 48 h; 5) deionized water, 30 min; 6) 3% Triton X-100 (BioChemica, Cambridge, MA, USA), 48 h; (7) PBS with 100 U/mL of penicillin and 100 μg/mL of streptomycin (working concentration 10 mg/L), 48 h. The solutions were changed once a day. Each solution was sterile. The perfusion was conducted at room temperature. The completeness of decellularization was controlled by histological study after staining with hematoxylin and eosin and alcian blue.

Lyophilization of decellularized tissue was carried out in 15-SRC-X sublimator (VirTis, Los Angeles, CA, USA); the process included freezing at a temperature of −80 °C followed by drying in a vacuum chamber for 48 h in a “floating” mode with a temperature transition from −40 to +20 °C; lyophilized cartilage was homogenized. The resulting granules were sieved through a stainless-steel sieve with a pore diameter of 280 μm (Vibrotechnik, Saint Petersburg, Russia). The sieved granules were stored at +4 °C in a sealed container in the dark. The size (diameter) of the granules was analyzed via microscopy (Biomed-3, Biomed, Moscow, Russia; UCMOS 14000KPA camera, ToupCam, Shanghai, China) and ImageJ 1.52a. The measured area of the granules was used to calculate its diameter (Equation (1)). A diameter frequency distribution plot was made in R version 4.1.2.
(1)$$D=2\times \sqrt{S\times \pi }$$where D—diameter of the granules, S—area of granule slice.

### 4.3. Bioink Preparation

The bioink was based on sterile type I pig atelocollagen. All the procedures with the material were in accordance with the previously described method [65]. Briefly, on the day of the experiment, the cells were removed from a culture flask with trypsin-EDTA solution. After staining with trypan blue, the number of living cells was counted; the cells were centrifuged (400 g, 5 min) and resuspended in 0.20 mL of serum-free medium. The dECM granules in an amount of 20 mg were added to 0.20 mL of collagen buffer solution. Then, the cell suspension in amount of 20 × 106 mL^−1^ was mixed (1:1) with the buffer solution (containing dECM). In turn, the resulting solution was mixed (1:1) with 8% collagen hydrogel. Thus, the main bioink option contained 40 mg/mL collagen (4% solution), 25 mg/mL dECM (2.5%) and 5 × 10^6^ mL^−1^ MSC. Before use, the bioink was kept at +4 °C. Three different compositions (formulations) of collagen-based bioink were investigated: (1) with dECM; (2) with MSC; (3) with dECM and MSC.

### 4.4. Bioprinting

Bioprinting was performed on Rokit Invivo 3D (Seoul, South Korea) using 1.68 firmware. The input printing model was sliced using NewCreatorK 1.57.63. The hydrogel was contained in a glass “Luer-Lock” type syringe with 8.24 mm inner diameter. The material was extruded through a 21G (inner diameter ~ 514 µm) with 0.5 in length. The printing was carried out into sterile 60 mm Petri dishes (Corning, New York, NY, USA), which were placed on a printing table. The temperature of the printing table, as well as the dispenser, was +4 °C.

In order to assess the printability of the bioink (with dECM), a test object with 8 niches (of 4 mm^2^ area each) was developed. The height of the object was 1000 μm, the height of the printing layer was 333 µm. Taking into account the potential error of the “0” layer (base level) calibration process (accuracy ±100 µm), the material yield on the first layer was 115% (Equation (2)). The testing was repeated twice.
(2)$$\frac{H+H_{1}}{H}\times 100\%=O_{1}$$where O_1_—material output on the first layer, %; H—print layer height; H_0_—average error of the base level estimation.

The scaffolds used for the implantation were cube-shaped with dimensions of 4 × 4 × 4 mm. The printing parameters were the same, except for the percentage of material output. This parameter corresponded to 150%. Other parameters were changed to maintain 100% material fill level. Before the printing, the printer chamber was sterilized using a built-in 254 nm UV-lamp. In order to induce collagen polymerization, immediately after, the printing warm DMEM medium was added to the scaffold. Before the implantation, the scaffolds were incubated for ~24 h under standard culture conditions.

### 4.5. Implantation in Animals

The scaffolds were implanted under the skin in the withers area in 12 outbred white male rats (155–230 g; aged 1.5–2 months). All operations with the animals were conducted under ether inhalation anesthesia. The animals’ hair was cut off in the withers area. The operating field was treated with 70% ethanol solution. An incision made with scissors and a scalpel was used to form a pocket under the skin where the implant was placed. In accordance with the study design, three groups were formed with four animals in each one: (1) collagen-based scaffolds containing only dECM granules; (2) collagen-based scaffolds containing only MSC; (3) collagen-based scaffolds containing dECM and MSC. The edges of the pocket were pulled together, the wound was sutured (Monocryl Poliglecaprone 25, Ethicon, Raritan, USA). The implantation site was marked with a colored thread of suture material. The seam was treated with 3% hydrogen peroxide solution. The second suture was applied to the skin and re-treated with hydrogen peroxide. A medical adhesive was applied on top to ensure better fixation. The area around the operating field was anesthetized with 0.5% novocaine. The surgical site was examined daily. On the 7th and 14th days, two animals from each group were euthanized; the material was taken for histological examination. All procedures at this stage were conducted according to the Bioethical Commission on Keeping and Using Laboratory Animals (No. 1-N-00007 dated 7 May 2021).

### 4.6. Histological and Immunohistochemical Studies

The scaffolds with surrounding tissue fragments were fixed for 24 h in an acidic Bouin solution (including 1.3% trinitrophenol (Sigma-Aldrich, St. Louis, MO, USA), 40% formalin (hereinafter BioVitrum (Moscow, Russia), unless otherwise specified)). After washing in 70% ethanol, standard histological preparation of the samples was performed; they were then placed in a paraffin medium (Histomix). Paraffined slices of 5 μm thick were obtained with a microtome (Leica RM2235) and placed on silanized glasses (S3003, Dako). Dewaxed slices were stained with hematoxylin and eosin, alcian blue (8GX, Sigma-Aldrich, St. Louis, MO, USA) for histological studies. The bone tissue was visualized using Masson's trichrome. The slices were dehydrated in alcohol, cleared in (ortho-)xylene, and embedded in Canadian balsam (Merck).

Polyclonal rabbit antibodies to proliferating cell nuclear antigen (PCNA, PA5-27214, 1:200, Invitrogen) and monoclonal rabbit antibodies to type II collagen (SAB4500366, 1:50, Sigma-Aldrich, St. Louis, MO, USA) were used for immunohistochemical studies. Secondary goat antibodies were conjugated with horseradish peroxidase (ab205718, 1:1000, Abcam). Immunohistochemical solutions were prepared in PBS. Following the immunohistochemical study protocol, dewaxed slices immersed in citrate buffer (pH 6.0) were boiled (5 min) before primary antibodies to PCNA and type II collagen were applied. Endogenous peroxidase was blocked in 3% hydrogen peroxide solution. The blocking buffer was supplemented with 2% serum of secondary antibody donors, 1% bovine serum albumin, and 0.1% Triton X-100. The samples were incubated in the primary antibody solution overnight in a humid chamber at +4 °C. After washing in PBS, secondary goat anti-rabbit antibodies were applied to the slices for 1 hour at room temperature. Substrate peroxidase was detected using diaminobenzidine (Liquid DAB+, K3468, Dako). Additional staining with hematoxylin was carried out for better visualization of tissue sections. The slices were dehydrated in alcohol, cleared in (ortho-)xylene, and embedded in Canadian balsam (Merck). Histological slices were examined by AXIO Imager A1 microscope (Carl Zeiss) and Canon Power Shot A640 camera (Canon).

### 4.7. Statistics

All calculations were performed in R 4.1.0. IQR was applied to the data on dECM granule diameter since its distribution was not normal (according to Shapiro–Wilk test). Additionally, the empirical cumulative distribution function was used to calculate the quantile for a certain diameter value. The data related to bioink printability were described with mean, standard error, and 95% confidence interval. Comparison of the formulations was carried with Welch’s *t*-test and confidence interval of the difference. The results of the ADSC phenotype study were described with mean and standard error.

## 5. Conclusions

In recent years, decellularized tissues have become a powerful platform for creating tissue scaffolds. ECM of each tissue creates a unique tissue-specific microenvironment for resident cells, providing them with the structure and biochemical signals necessary for their functioning. In our study, we showed that dECM granules (up to 280 µm) added to the collagen-based bioink could stimulate ADSC differentiation in the chondrogenic direction under in vivo conditions. The effect was more pronounced in the case of MSC-laden scaffolds (using 3-component bioink) and less evident for cell-free scaffolds (2-component, cell-free bioink). At the same time, further development of the technique requires solving the problems related to scaffold vascularization, as well as the influence of the mechanical properties of the ready scaffolds on MSC differentiation.

## Figures and Tables

**Figure 1 ijms-23-02703-f001:**
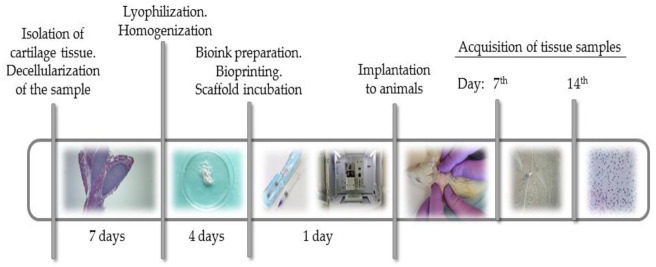
Study design. dECM powder was obtained from rat costal cartilage by decellularization, lyophilization and milling. Bioink containing sterile type I pig atelocollagen, dECM powder and human MSC were used to print scaffolds. The scaffolds were implanted under the skin in the withers area in 12 rats. The histological and immunohistochemical studies of material from animals was done on days 7 and 14 after implantation.

**Figure 2 ijms-23-02703-f002:**
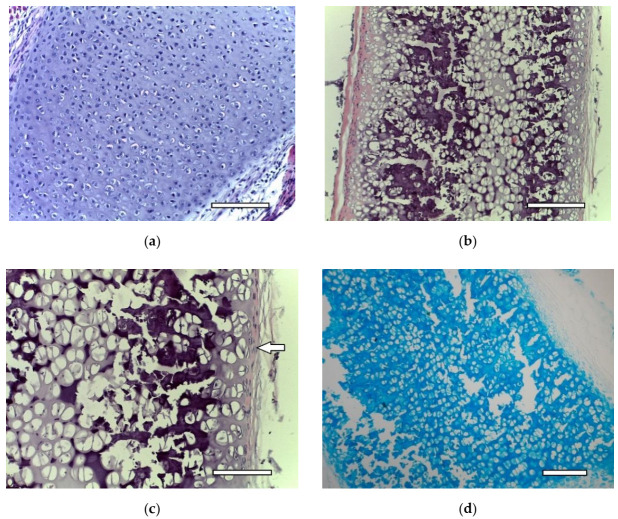
Costal rat cartilage. (**a**) native cartilage, staining with hematoxylin and eosin, objective lens ×20, scale bar—100 μm; (**b**) decellularized sample, staining with hematoxylin and eosin, objective lens ×10, scale bar—200 μm; (**c**) decellularized sample, staining with hematoxylin and eosin, objective lens ×20, scale bar—100 μm. Perichondrium cells are shown by arrows; (**d**) decellularized sample, staining with alcian blue, objective lens ×10, scale bar—200 μm.

**Figure 3 ijms-23-02703-f003:**
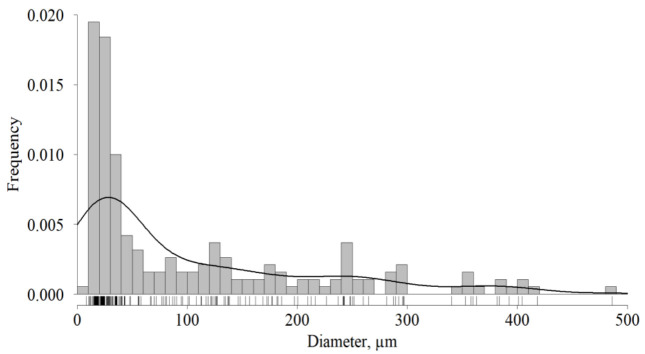
Frequency distribution of dECM granules diameter, *n* = 188.

**Figure 4 ijms-23-02703-f004:**
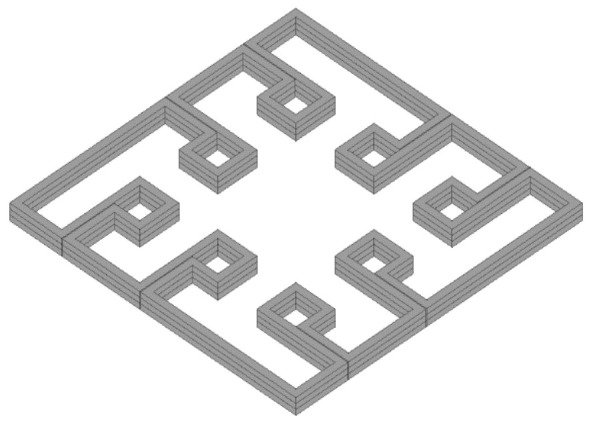
Input model of test printing bioink printability assessment.

**Figure 5 ijms-23-02703-f005:**
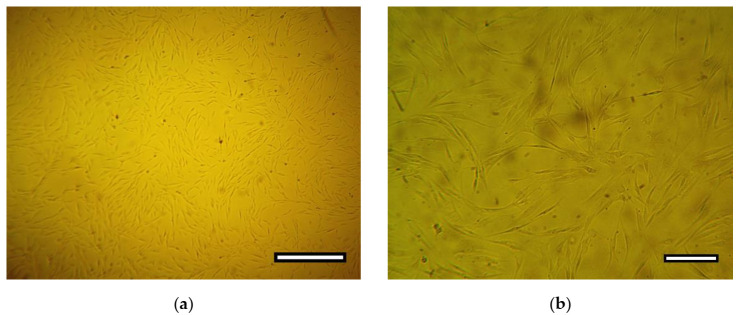
ADSC after 48 h of cultivation. Phase contrast: (**a**) objective lens ×4, scale bar—800 μm; (**b**) objective lens ×10, scale bar—200 μm.

**Figure 6 ijms-23-02703-f006:**
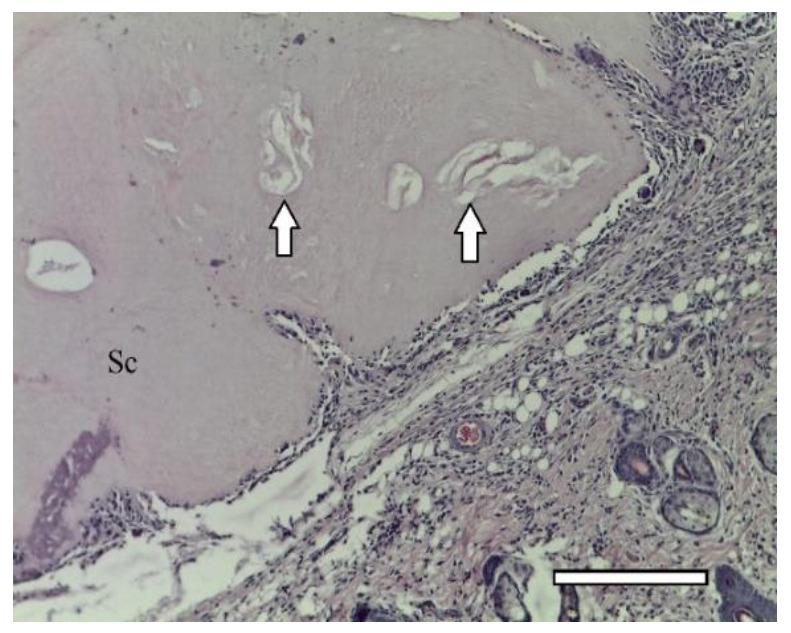
1st group (dECM only), 1 week. A connective tissue capsule surrounding the scaffold (Sc). Staining with hematoxylin and eosin, objective lens ×10, scale bar—200 μm. Cavities filled with fibrillar material are shown by arrows.

**Figure 7 ijms-23-02703-f007:**
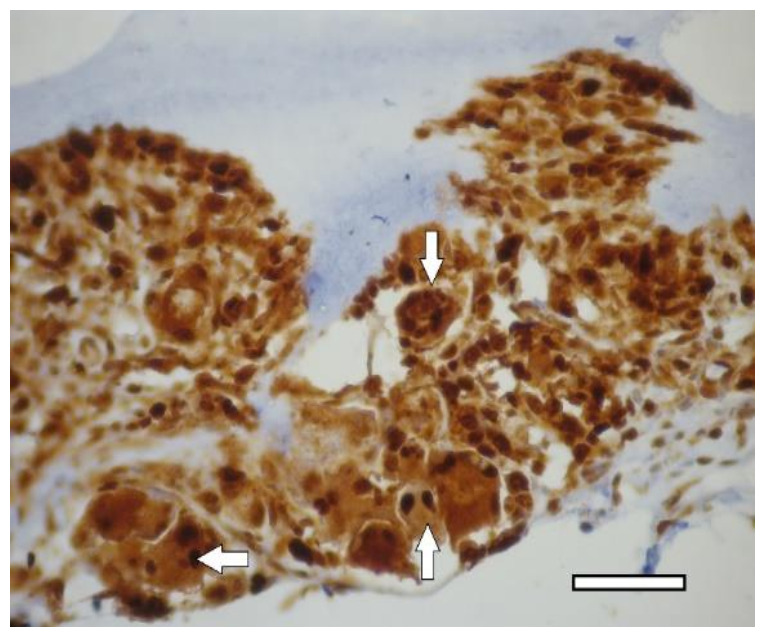
1st group (dECM only), 1 week. PCNA-positive multinucleated cells resorb around the perimeter of the scaffold in the connective tissue capsule (arrows). Immunohistochemical staining for PCNA, objective lens ×40, scale bar—50 μm.

**Figure 8 ijms-23-02703-f008:**
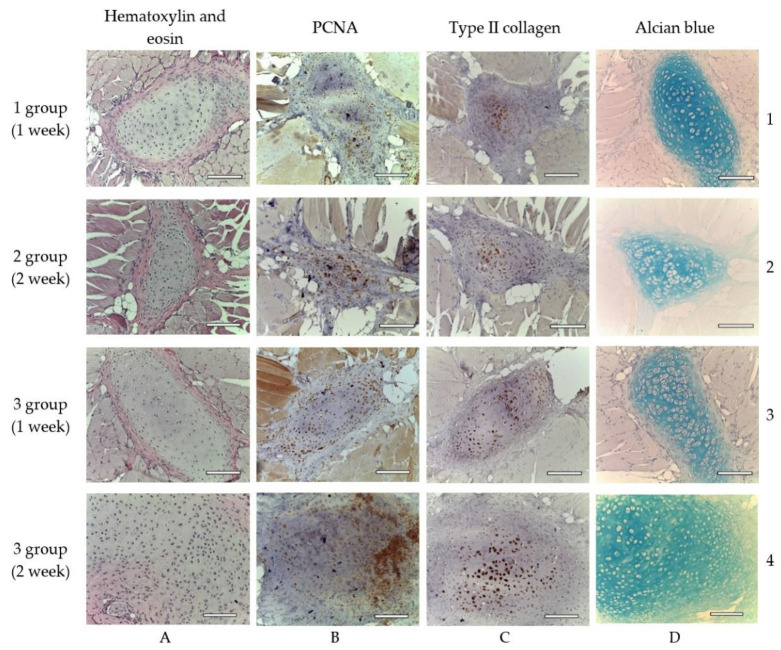
Cartilage tissue in animals of different experimental groups, objective lens ×20, scale bar—100 μm. Explanation in the text. (**A1**–**D1**)—1st group (dECM only), 1 week; (**A2**–**D2**)—2nd group (MSC only), 2 weeks; (**A3**–**D3**)—3rd group (dECM and MSC), 1 week; (**A4**–**D4**)—3rd group (dECM and MSC), 2 weeks. (**A**)—staining with hematoxylin and eosin; (**B**)—staining for PCNA; (**C**)—immunohistochemical staining for type II collagen; (**D**)—staining with alcian blue.

**Figure 9 ijms-23-02703-f009:**
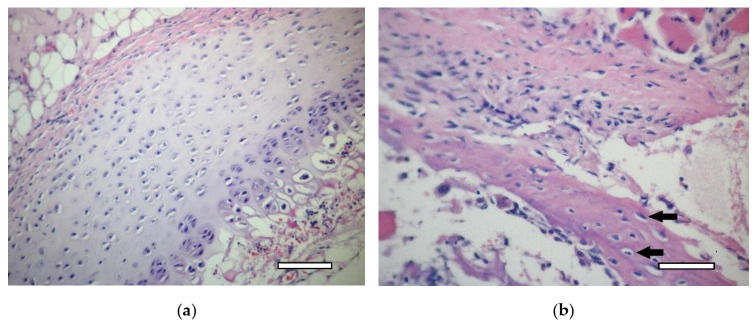
1st group (dECM only), 1 week. (**a**) cartilage tissue. Staining with hematoxylin and eosin, objective lens ×20, scale bar—100 μm; (**b**) formation of an osteoid with randomly located osteoblasts (arrows) in muscle tissue near the scaffold. Staining with hematoxylin and eosin, objective lens ×40, scale bar—50 μm.

**Figure 10 ijms-23-02703-f010:**
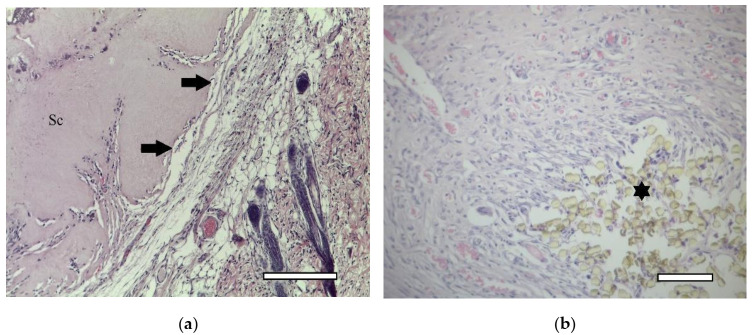
1st group (dECM only), 2 weeks. Connective tissue capsule (arrows) around the scaffold (Sc). Staining with hematoxylin and eosin. (**a**) objective lens ×10, scale bar—200 μm; (**b**) objective lens ×20, scale bar—100 μm. In the lower right corner, the remains of suture material, which marked the site of the implantation (asterics).

**Figure 11 ijms-23-02703-f011:**
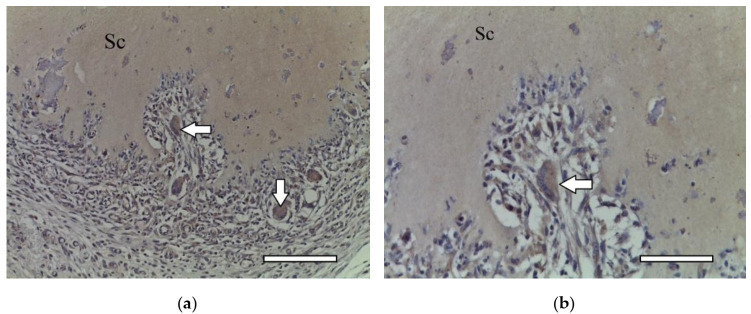
1st group (dECM only), 2 weeks. Collagen type II in the cytoplasm of multinucleated cells of foreign body resorption (arrows) in the connective tissue capsule. Diffuse background—scaffold material (Sc). Staining for type II collagen (asterisk). (**a**) objective lens ×20, scale bar—100 μm; (**b**) objective lens ×40, scale bar—50 μm.

**Figure 12 ijms-23-02703-f012:**
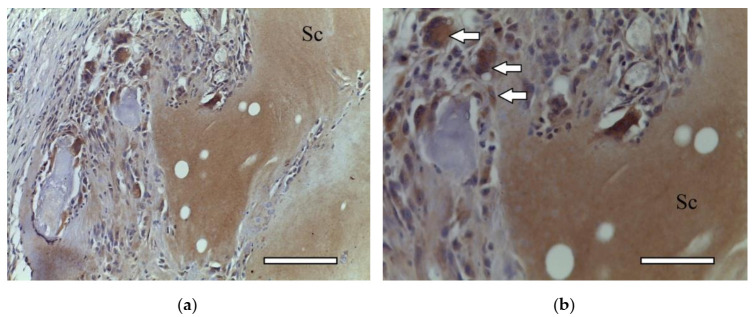
1st group (dECM only), 2 weeks. Collagen type II in the cytoplasm of multinucleated cells of foreign body resorption and macrophages (arrows) in the connective tissue capsule. Diffuse background—scaffold material (Sc). Staining for type II collagen (asterisk). (**a**) objective lens ×20, scale bar—100 μm; (**b**) objective lens ×40, scale bar—50 μm.

**Figure 13 ijms-23-02703-f013:**
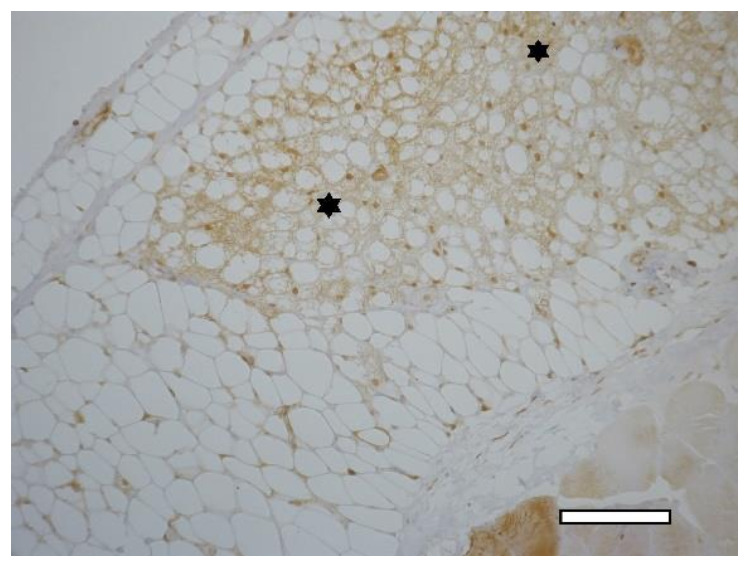
1st group (dECM only), 2 weeks. Staining of white and brown (asterisk) adipose tissue for nuclear antigen of proliferating cells, objective lens ×20, scale bar—100 μm.

**Figure 14 ijms-23-02703-f014:**
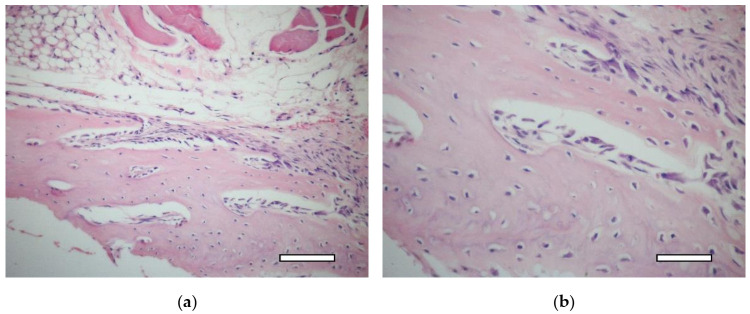
2nd group (MSC only), 1 week. Osteoid formation with randomly located osteoblasts near the implant. Staining with hematoxylin and eosin. (**a**) objective lens ×20, scale bar—100 μm; (**b**) objective lens ×40, scale bar—50 μm.

**Figure 15 ijms-23-02703-f015:**
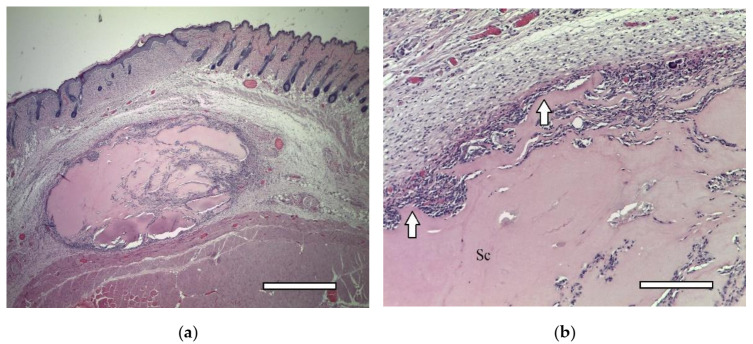
2nd group (MSC only), 1 week. Connective tissue capsule (arrows) around the implant. Staining with hematoxylin and eosin. Scaffold region (Sc). (**a**) objective lens ×2.5, scale bar—800 μm; (**b**) objective lens ×10, scale bar—200 μm.

**Figure 16 ijms-23-02703-f016:**
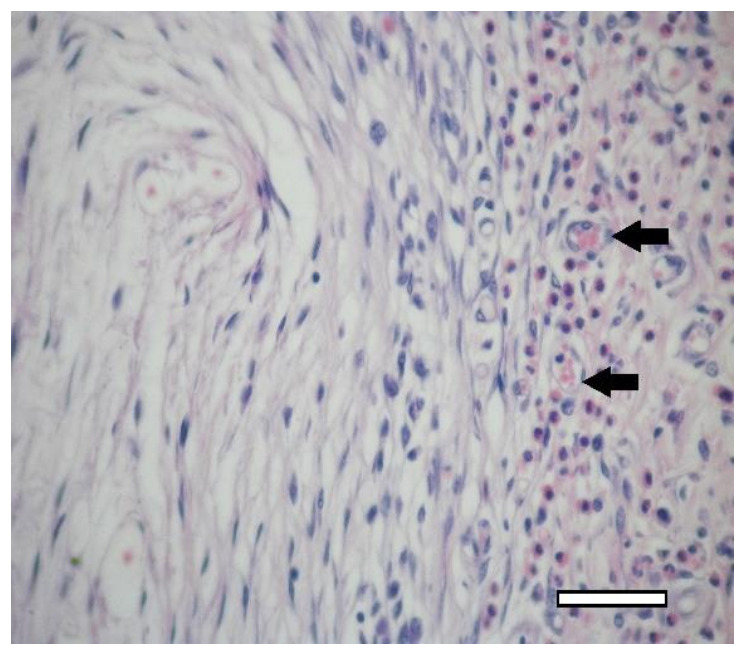
2nd group (MSC only), 2 weeks. Eosinophilic infiltrate (arrows) in the connective tissue capsule. Staining with hematoxylin and eosin, objective lens ×40, scale bar—50 μm.

**Figure 17 ijms-23-02703-f017:**
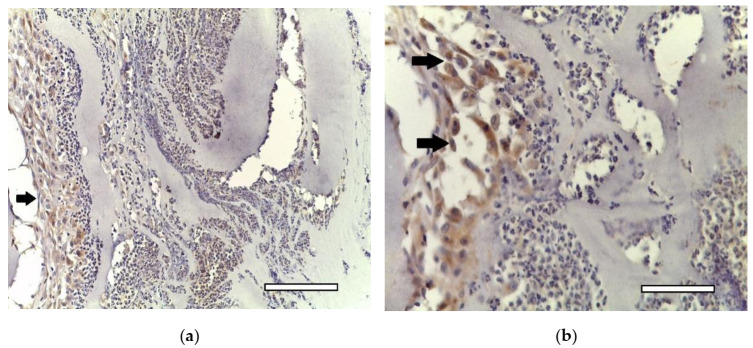
2nd group (MSC only), 2 weeks. Connective tissue capsule (arrows). Collagen type II staining. Macrophages with cytoplasm stained for type 2 collagen. (**a**) objective lens ×20, scale bar—100 μm; (**b**) objective lens ×40, scale bar—50 μm.

**Figure 18 ijms-23-02703-f018:**
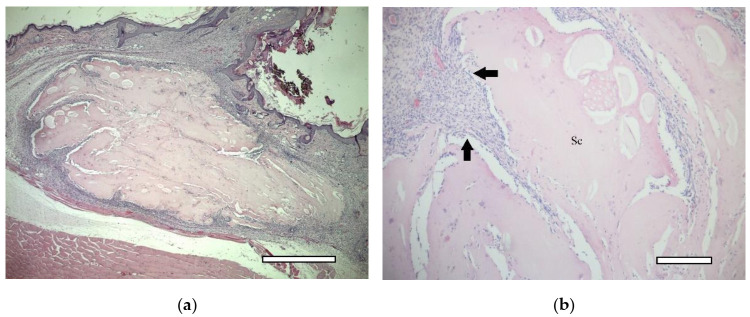
3rd group (dECM and MSC), 1 week. Connective tissue capsule. Staining with hematoxylin and eosin. Scaffold region (Sc) (**a**) objective lens ×2.5, scale bar—800 μm; (**b**) objective lens ×10, scale bar—200 μm.

**Figure 19 ijms-23-02703-f019:**
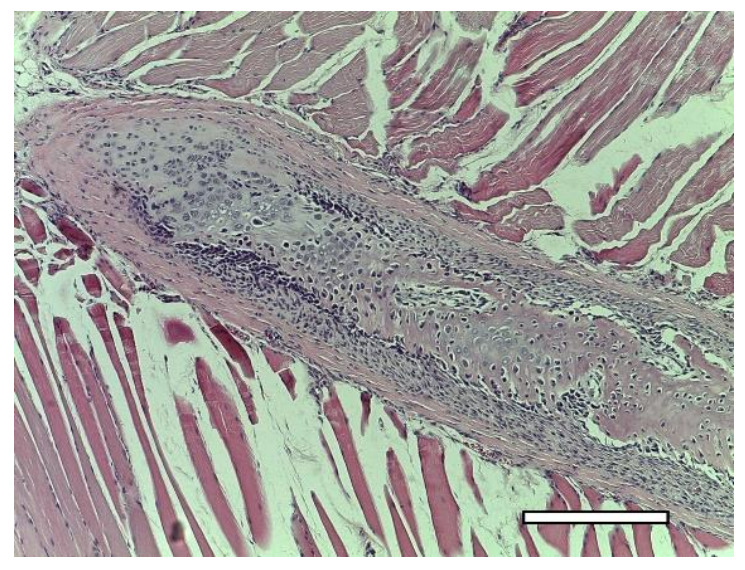
3rd group (dECM and MSC), 1 week. Cartilage cells with signs of indirect osteogenesis in the muscle tissue near the scaffold. Staining with hematoxylin and eosin. Objective lens ×2.5, scale bar—800 μm.

**Figure 20 ijms-23-02703-f020:**
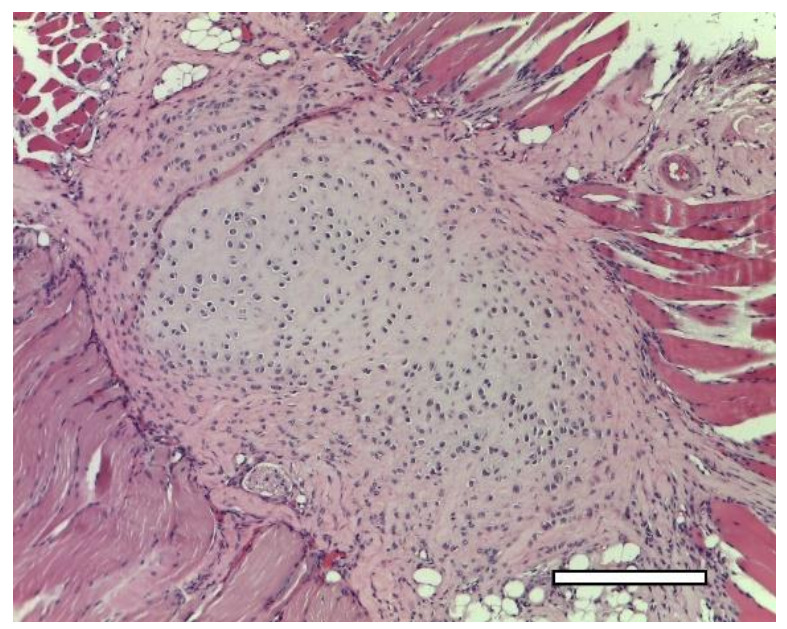
3rd group (dECM and MSC), 2 weeks. Cartilage tissue in the muscles near the scaffold. Staining with hematoxylin and eosin, objective lens ×10, scale bar—200 μm.

**Figure 21 ijms-23-02703-f021:**
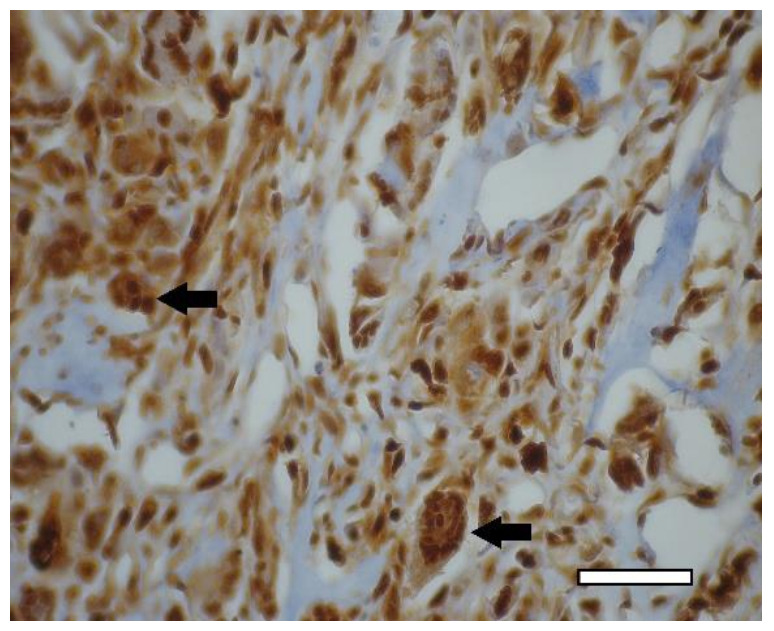
3rd group (dECM and MSC), 2 weeks. Immunohistochemical staining (PCNA) of multinucleated cells of cartilage tissue resorption (arrows), objective lens ×40, scale bar—50 μm.

**Table 1 ijms-23-02703-t001:** Results of test printing with 4% collagen-based collagen with and without dECM granules.

	Collagen		Collagen + dECM	
	*X* ± *S* (CI, 95%, *n* = 24)			
	After Printing	After Incubation	After Printing	After Incubation
Niches, mm^2^	1.516 ± 0.482 (1.249 ÷ 1.783)	1.286 ± 0.504 (1.017 ÷ 1.554)	1.411 ± 0.412 (1.192 ÷ 1.631)	1.097 ± 0.375 * (0.897 ÷ 1.297)
Lines, mm	0.886 ± 0.083 (0.851 ÷ 0.921)	0.920 ± 0.187 (0.845 ÷ 0.995)	0.901 ± 0.131 (0.846 ÷ 0.957)	0.957 ± 0.130 (0.902 ÷ 1.012)

*—Statistically significant difference according to Welch’s *t*-test in comparison with collagen after incubation.

**Table 2 ijms-23-02703-t002:** Immunophenotyping of MSC isolated from adipose tissue (*n* = 10).

Antibody	MSC Affinity	*X* ± *S_X_, %*
CD13	+	96.32 ± 1.08
CD 34	−	0.85 ± 0.22
CD 44	+	96.23 ± 1.37
CD 45	−	0.57 ± 0.21
CD 73	+	98.64 ± 0.59
CD 90	+	99.02 ± 0.77
CD 105	+	99.14 ± 0.36
CD 146	−	0.31 ± 0.12

## Data Availability

The study did not report any data.

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
