# Peer review of "Cartilage Formation In Vivo Using High Concentration Collagen-Based Bioink with MSC and Decellularized ECM Granules"

_ijms, 2022, doi:10.3390/ijms23052703_

Round 1
Reviewer 1 Report
The present paper is well written, designed and performed. The experiments were performed in a rigorous manner following the high methodological criteria.
I suggest to the authors to add in a supplementary materials the description and charcaterization of cells used in the study, with the morphological evaluation. In the introduction section authors should describe the importance of the use of decellularized extracellular matrix in the regenerative medicine (for e.g. doi.org/10.3390/biomedicines10020403).
Author Response
Reviewer 1
Dear Reviewer,
Thank you very much for your valuable suggestions and comments. We have revised our response, as well as the manuscript. We hope we were able to reasonably answer all your comments.
Yours faithfully,
Research Team
Revised response to the comments:
I suggest to the authors to add in a supplementary materials the description and charcaterization of cells used in the study, with the morphological evaluation
Morphological characteristics of cells with drawings are added to the main text of the article in the section Results 2.4. ADSC phenotype (Fig.5): «The cells had a regular fusiform shape with 2-4 long processes, the cytoplasm was homogeneous and transparent, without any inclusions. The nuclei were located closer to the periphery (eccentrically) with a uniform distribution of chromatin. (Fig.5) The cell culture had a high proliferative potential. The cell population doubling time was 24.31±2.19 h.»
In the introduction section authors should describe the importance of the use of decellularized extracellular matrix in the regenerative medicine (for e.g. doi.org/10.3390/biomedicines10020403).
In the introduction added a sentence with a link to the article: «The importance of using biomaterials, including dECMs, in regenerative medicine can hardly be overestimated, since they allow you to restore the function of the organ, and not just its volume. So, in the study [25]developed a decellularized dental pulp (DDP) matrix loaded by human Dental Pulp Stem Cells (hDPSCs) in co-treatment with Extracellular Vesicles (EVs), which might enhance the dentinogenic differentiation with a high potentiality for endodontic regeneration.»
Reviewer 2 Report
Currently, the article has several weaknesses and I suggest that authors should improve it in order to resubmit it. I recommend enhancing the wording (if processes are listed, continue with the numbering throughout the text ( i. e. line 45: 1); indicate correctly the exponentials in the numbers); improving the introduction as well as the discussion and providing rheological characterisation of bioinks, images of the scaffold microstructure (SEM), immunostaining, among others.
Although the idea of the article is interesting, from my point of view the article should be strongly improved in order to be accepted in the journal. On the one hand, the wording should be improved. Throughout the article, they report the formation of cartilage in the areas surrounding the implanted scaffold, so the title might be confusing.
Regarding the experimental section, since they talk about the relevance of the rheological properties of the hydrogel, I miss the evaluation of these properties as well as the microscopic image (SEM) of the internal structure of these hydrogels.
In line 318, they report that their scaffold has an allogenic nature while in the Material and Methods section, they describe that one of the components of the bioink is porcine collagen (line 461: “The bioink was based on sterile type I pig atelocollagen”). Although freeze-drying of the dECM is one of their fundamental processes for the manufacture of the granules, they also do not report the process conditions.
Images of fluorescent immunostaining with more specific markers are missing. The current histological images, I do not have enough knowledge to assess whether what they indicate is correct. I would recommend delimiting the scaffold area in the images to facilitate interpretation by the reader, as well as grouping several of them in the same image mosaic.
With my tools, I am not able to detect plagiarism. After the collection of all these suggestions, I consider that the article should either be strongly improved for publication or rejected.
Author Response
Reviewer 2
Dear Reviewer,
Thank you very much for your valuable suggestions and comments. We have revised our response, as well as the manuscript. We hope we were able to reasonably answer all your comments.
Yours faithfully,
Research Team
Revised response to the comments:
I recommend enhancing the wording (if processes are listed, continue with the numbering throughout the text ( i. e. line 45: 1)
section Iintroduction: The text was сorrected
Improving the introduction as well as the discussion and providing rheological characterisation of bioinks, images of the scaffold microstructure (SEM)
We agree with this remark and made the necessary changes to the manuscript, since in this study we did not study the rheological properties of bioink, the component of which is dECM powder. The main objective of this study was to elucidate the possibility of MSCs differentiation in the chondrogenic direction under the influence of only signal molecules of the extracellular matrix without the use of additional stimulants. The rheological properties of bioink based on only collagen without powder were studied by our co-authors [https://doi.org/10.1007/s10856-019-6233-y].
Additional rheological studies, mechanical and scaffold microstructure properties are our next work for this research field, we hope the next work will bring interesting results for readers.
Throughout the article, they report the formation of cartilage in the areas surrounding the implanted scaffold, so the title might be confusing.
We would like to leave the title of the article unchanged, since both MSCs and dECM granules were delivered to animals as part of collagen-based bioink.
In line 318, they report that their scaffold has an allogenic nature while in the Material and Methods section, they describe that one of the components of the bioink is porcine collagen (line 461: “The bioink was based on sterile type I pig atelocollagen”).
The bioink from which the scaffolds were obtained included several components: porcine atelocollagen and human MSCs, or porcine atelocollagen and dECM powder from rat rib cartilage, or porcine atelocollagen, human MSC and dECM powder from rat rib cartilage. The xenogenic components include porcine atelocollagen and human MSCs, the matrix powder was obtained from rats and administered to rats, so the source of the matrix is allogeneic. Thus, not the entire framework is allogeneic, but one of its components. We wrote about this in the Discussion, there is no contradiction here.
Although freeze-drying of the dECM is one of their fundamental processes for the manufacture of the granules, they also do not report the process conditions.
Required addition added to text Section Materials and Methods 4.2. dECM granules:
«…the process included freezing at a temperature of -80°C followed by drying in a vacuum chamber for 48 hours in a "floating" mode with a temperature transition from -40 to +20C»
Images of fluorescent immunostaining with more specific markers are missing
Specific markers of cartilage tissue are type II collagen synthesized by chondrocytes and glycosaminoglycans (GAGs). Antibodies labeled with both fluorescent dyes and horseradish peroxidase (either one or the other) are used to detect collagen. GAGs are detected by staining histological sections with alcian blue in an acidic environment. They cannot be detected by immunostaining, since they are not proteins and do not enter into immunological reactions. Thus, the used methods for indicating specific products formed by cartilage tissue cells are sufficient and do not need additional confirmation.
I would recommend delimiting the scaffold area in the images to facilitate interpretation by the reader, as well as grouping several of them in the same image mosaic.
We have added the necessary notation. On those images where there is a scaffold, its area is marked with the letters Sc (short for Scaffold, given in the figure captions). (Fig. 6, 10, 11, 12, 15, 19).
Round 2
Reviewer 1 Report
Authors adressed to all my comments.
Author Response
Dear Reviewer,
Thank you very much for your valuable suggestions and comments.
Yours faithfully,
Research Team
Reviewer 2 Report
Thank you for the modifications made to the manuscript following the suggestions previously indicated.
Although the conclusion section is not mandatory, it is suggested to add this part to summarize the long discussion section and clearly highlight the importance of this research.
Regarding the format, I would again recomend grouping some images in the same figure for smoother reading.
Author Response
Dear Reviewer,
Thank you very much for your valuable suggestions and comments. We have revised our response, as well as the manuscript. We hope we were able to reasonably answer all your comments.
Yours faithfully,
Research Team
Revised response to the comments:
Reviewer 2
Although the conclusion section is not mandatory, it is suggested to add this part to summarize the long discussion section and clearly highlight the importance of this research.
Regarding the format, I would again recomend grouping some images in the same figure for smoother reading.
We added conclusion section to the manuscript. We also modified figures. Where it was possible, we combined them into collages
«Conclusion» section added to manuscript:
«In recent years, decellularized tissues have become a powerful platform for creating tissue scaffolds.ECM of each tissue creates a unique tissue-specific microenvironment for resident cells, providing themwith the structure and biochemical signals necessary for their functioning. In our study, we showed that dECM granules (up to 280 µm) added to the collagen-based bioink could stimulate ADSC differentiation in the chondrogenic direction under in vivo conditions. The effect was more pronounced in the case of MSC-laden scaffolds (using 3-component bioink) and less evident for cell-free scaffolds (2-component, cell-free bioink). At the same time, further development of the technique requires solving the problems related to scaffold vascularization, as well as the influence of the mechanical properties of the ready scaffolds on MSC differentiation».